# Effect of Subconjunctival Healaflow in Filtrating Surgery with Preserflo MicroShunt in Primary Open Angle Glaucoma

**DOI:** 10.3390/jcm14176000

**Published:** 2025-08-25

**Authors:** Leonie F. Keidel, Miranda Gehrke, Stefan Kassumeh, Lara Buhl, Siegfried G. Priglinger, Marc J. Mackert

**Affiliations:** Department of Ophthalmology, LMU University Hospital, LMU Munich, Mathildenstr. 8, 80336 Munich, Germany; miranda.gehrke@med.uni-muenchen.de (M.G.); stefan.kassumeh@med.uni-muenchen.de (S.K.); lara.buhl@med.uni-muenchen.de (L.B.); siegfried.priglinger@med.uni-muenchen.de (S.G.P.); marc.mackert@med.uni-muenchen.de (M.J.M.)

**Keywords:** Healaflow, filtrating surgery, primary open angle glaucoma, PreserFlo^®^ MicroShunt

## Abstract

**Background/Objectives**: This work aims to clarify whether the subtenon use of the sodium hyaluronate product Healaflow in filtrating surgery with PreserFlo MicroShunt positively influences the early postoperative course in terms of control of intraocular pressure, hypotony, and needling rate. **Methods**: A retrospective, randomized controlled, interventional, single-center trial was performed at the Ludwig Maximilians-University Munich from January 2024 to July 2024. Only patients with primary open angle glaucoma (POAG) were included. In all patients, a complete ophthalmological examination including best corrected visual acuity (BCVA), automated refraction, and Goldman tonometry was performed at 2 days, 1–4 and 5–8 weeks, and 3–4 and 5–6 months after surgery. Healaflow was injected underneath the tenon during filtrating surgery with PreserFlo MicroShunt in addition to mitomycin C (MMC). The Healaflow group was compared to a control group with POAG patients in which Healaflow was not used during surgery with PreserFlo MicroShunt and MMC. **Results**: A total of 45 eyes of 45 patients were included, with 20 eyes in the Healaflow group and 25 eyes in the control group. In both groups, a significant reduction in IOP and medication could be observed: complete surgical success (IOP ≥ 6 mm Hg and ≤17 mm Hg, without surgical complications or complete loss of vision) was reached in 88% of patients in the Healaflow group at the last follow-up. In 95% of patients in the control group, complete success could be observed. The success rates did not significantly differ between the two groups (*p* = 0.568). Hypotony rates were 35% in the Healaflow and 12% in the control group after two days (*p* = 0.083); the rates equalized after 1–4 weeks (*p* = 1). Needling rates were comparable between both groups (25% versus 20%, *p* = 0.731). **Conclusions**: PreserFlo MicroShunt implantation with MMC was equally effective in terms of reduction in IOP and medication in both scenarios with additional or without the use of Healaflow. Postoperative hypotony and needling rates did not significantly differ between the two groups. The additional effects of Healaflow on anti-scarring and maintaining space are likely too minimal to cause significant differences in IOP or medication when already treated with MMC.

## 1. Introduction

Currently, a filtrating procedure using a shunt implantation (PreserFlo^®^ MicroShunt, Santen Pharmaceutical Co., Ltd., Osaka, Japan) is used to treat moderate to severe glaucoma [1,2]. This procedure causes an immediate reduction in intraocular pressure, but the bleb is, like in any other filtrating surgery, prone to tissue remodeling and scarring. An inflammatory reaction in the region of the bleb that can potentially lead to filtration failure and the need for surgical needling if the intraocular pressure increases [3].

Healaflow^®^ (Anteis S.A., Geneva, Switzerland) is a novel cross-linked and non-cross-linked sodium hyaluronate product that is used to preserve the function of a filtering bleb by maintaining the volume of the subconjunctival filtration space. The structural composition is designed to contribute to maintaining a steady and uniform flow of the aqueous humor. Moreover, it is slowly absorbed by the tissue and is designed to inhibit tissue remodeling and scarring reaction after glaucoma surgery due to its biological properties [4,5,6]. In contrast to the already-established mitomycin C (MMC), it does not act cytotoxically and is more targeted towards preventing the accumulation of fibroblasts and collagen [7]. The viscoelastic implant can be applied underneath the sclera in non-penetrating glaucoma surgery or underneath the tenon during the operation.

To date, relevant data on Healaflow^®^ are available in the context of implant-free glaucoma surgery, such as trabeculectomy or deep sclerectomy. In these studies, Healaflow^®^ was injected underneath the flap or subtenon to create a well-functioning bleb while inhibiting angiogenesis and reducing inflammation. The current data revealed that the injection of Healaflow^®^ was safe and effective [4,5,6,8,9,10] with a low rate of complications [7]: Wang et al. reported that although patients of both standalone trabeculectomy and trabeculectomy combined with Healaflow^®^ experienced significant reductions in intraocular pressure, patients treated with Healaflow^®^ showed greater and statistically significant IOP reduction at 1, 3, and 5 years post-operation [9]. Feusier and Roy et al. investigated the use of Healaflow in deep sclerectomy and found that the use of Healaflow also reduced the need for goniopuncture and needling of the filtering bleb [4].

There are currently no clinical data of the effect of Healaflow^®^ during filtrating surgery with PreserFlo^®^ MicroShunt.

The aim of this study is to retrospectively evaluate the early postoperative course after PreserFlo^®^ MicroShunt implantation with MMC and additional use of subtenon Healaflow^®^ in patients with primary open glaucoma (POAG). This group will be compared to a control group that underwent PreserFlo^®^ MicroShunt surgery without the use of Healaflow^®^.

## 2. Materials and Methods

Retrospective monocentric analysis of the early postoperative course of patients with POAG who underwent filtrating surgery using PreserFlo^®^ MicroShunt (Santen Pharmaceutical Co., Ltd., Osaka, Japan) with the use of Healaflow^®^ (Anteis S.A., Plan-les-Ouates, Switzerland) with group comparison with a POAG control collective operated on using Preserflo™ MicroShunt without Healaflow^®^.

For this retrospective case–control study, 20 consecutive POAG patients who underwent PreserFlo^®^ MicroShunt filtrating surgery with the use of Healaflow at the Hospital of the Ludwig Maximilians-University Munich, Department of Ophthalmology, Germany from January to June 2024 were included in this study. This group was matched by age, spherical equivalent, glaucoma severity (mean deviation on visual field testing, Humphrey Field Analyzer), and baseline intraocular pressure to a POAG control group (25 eyes) who underwent PreserFlo^®^ MicroShunt filtrating surgery without the use of Healaflow from January 2024 to June 2024 (Table 1). Inclusion criteria included patients with moderate to severe POAG undergoing surgery with PreserFlo^®^ MicroShunt due to inadequate intraocular pressure despite maximum local pressure-lowering therapy, with eye drop intolerance and/or progression on visual field testing, with or without previous (MIGS) surgery. Exclusion criteria include combined procedure with cataract surgery, silicone oil tamponade, partner eye already included in the study, age younger than 18, and selective laser trabeculoplasty within the last 90 days preoperatively. Patients were recruited via the glaucoma outpatient department.

The study was approved by the ethics committee, with the identifier 25-0296, as a retrospective case–control analysis. The study adhered to the tenets of the Declaration of Helsinki.

Epidemiological data were collected for each patient, including age, gender, history of ocular comorbidities or procedures, and objective refraction-based visual acuity measured with the Snellen chart, which was subsequently converted to logMAR for analysis. Spherical equivalent values were obtained using the Nidek AR-1s auto refractometer (Oculus GmbH, Wetzlar, Germany). Intraocular pressure was obtained using Goldmann applanation tonometry. These examinations were performed 2 days, 1–4 weeks, 5–6 weeks, 3–4 months, and 5–6 months after surgery. Additionally, standard automated perimetry was conducted using a Humphrey Field Analyzer (Carl Zeiss AG, Oberkochen, Germany) prior to surgery. Due to the retrospective study design, outcome assessors were not blinded for the type of surgery.

Surgical success was defined as achieving an intraocular pressure between 6 mmHg and 17 mmHg at the final follow-up visit, together with a reduction of at least 20% from baseline IOP, with or without the use of topical antiglaucomatous medications. Success further required the absence of oral carbonic anhydrase inhibitor use, no need for surgical revision or additional glaucoma surgery, and preservation of light perception.

All patients received filtrating surgery with the use of PreserFlo^®^ MicroShunt according to the manufacturer’s instructions for use by one glaucoma surgeon (M.M.). Either a retrobulbar anesthesia with 4% lidocaine and 0.5% bupivacaine or a general anesthesia was used. Subtenon Mitomycin C (0.02 mg/mL for 2 min) was used in all patients. After preparation of the bleb and PreserFlo^®^ MicroShunt insertion, 20 patients received an injection of Healaflow^®^ (Anteis S.A., Geneva, Switzerland, 2.25%) using a 25 G needle (0.2–0.5 mL) underneath the tenon before conjunctival closure to maintain bleb elevation according to standard protocols and the manufacturer’s instructions.

Following surgery, 4 mg of dexamethasone was administered peribulbarly. In both groups, 1% prednisolone acetate eye drops were applied hourly during the first week, then gradually tapered over six weeks. Postoperative interventions—including subconjunctival 5-FU (0.1 mL) injections, needling combined with 5-FU, surgical revisions, or reinitiation of topical IOP-lowering therapy—were performed at the discretion of the treating ophthalmologist.

### Statistical Analysis

Data collection and initial analysis were conducted using Microsoft Excel (Version 16.23 for Mac; Microsoft, Redmond, WA, USA). Statistical analyses were performed with SPSS Statistics 26 (IBM Germany GmbH, Ehningen, Germany), with a significance threshold set at *p* < 0.05. The Kolmogorov–Smirnov test was used to assess normality of data distribution. Inter-group comparisons were conducted using independent and paired two-tailed Student’s t-tests for normally distributed variables (e.g., preoperative IOP) and the Mann–Whitney U test for non-normally distributed variables (e.g., age at time of surgery, preoperative MD). For comparisons involving more than two groups, repeated-measures ANOVA with Geisser–Greenhouse correction was applied. Graphs were generated using GraphPad Prism 9 (GraphPad Software, San Diego, CA, USA). To assess the adequacy of the sample size, a post hoc power analysis was performed based on the observed data.

## 3. Results

### 3.1. Patient Characteristics

In total, 45 eyes of 45 POAG patients were included: 20 eyes received filtrating surgery with PreserFlo^®^ MicroShunt and the use of subtenon Healaflow (Healaflow^®^ group), while 25 eyes received a PreserFlo^®^ MicroShunt implantation without using Healaflow^®^ (control group). The groups were matched for age, SE, baseline IOP, and severity of glaucoma. Demographic characteristics, SE, and perimetry findings (MD) for the two groups can be seen in Table 1. All patients were white. Regarding previous glaucoma surgeries/laser treatments (SLT), eight patients (40%) underwent previous procedures in the Healaflow^®^ and four (16%) in the control group (*p* = 0.096, Table 1).

### 3.2. Development of IOP

Both groups showed a significant reduction in IOP at the endpoint 4–6 months after surgery: In the Healaflow^®^ group, an IOP reduction to 13.5 ± 4.1 mmHg, *p* = 0.037 could be observed. The control group showed an IOP reduction to 11.42 ± 3.0 mmHg, *p* = 0.015 (Figure 1). The difference in IOP between the two groups was not significant at any time point (1d: −0.62 mmHg (95% CI: −4.37 to 3.13), *p* > 0.999, 2d: 0.46 mmHg (95% CI: −3.10 to 4.02), *p* = 0.997, 1–4 weeks: 0.35 mmHg (95% CI: −3.57 to 4.27) *p* = 0.851, 4–8 weeks: 0.40 mmHg (95% CI: −1.63 to 2.43 mmHg), *p* = 0.999, 3–4 months: −1.78 mmHg (−4.26 to 0.70), *p* = 0.997, 4–6 months: −2.08 (95% CI: −4.36 to 0.20 mmHg), *p* = 0.132), as shown in Figure 1. Given the small sample size and non-significant result, a post hoc power analysis was conducted to assess the likelihood of detecting the observed effect for each of the different time points. The achieved powers were calculated to be 0.062 at day 1, 0.057 at day 2, 0.710 at weeks 1–4, 0.068 at weeks 5–8, 0.293 at months 3–4, and 0.437 at months 5–6.

In both groups, a significant reduction in medication could be observed. At the endpoint 4–6 months after surgery, patients in the Healaflow^®^ group required 0.25 ± 0.6 (*p* < 0.0001) medications (preoperative number of medications: 2.90 ± 0.81 (1–5)), while patients in the control group required 0 ± 0 (*p* < 0.0001) (preoperative number of medications: 2.79 ± 1.44 (0–6), Figure 2). The difference in the reduction in medication between the two groups was not statistically significant (−0.25, 95% CI −0.54 to 0.04, *p* = 0.118). The achieved power was calculated to be 0.74.

A total of 88% of patients in the Healaflow^®^ group showed complete success; two patients had an intraocular pressure >17 mmHg at the last follow up visit under medication. One of these patients required open revision to reach the target pressure, while the other patient required needling and an additional cyclodiode treatment to reach the target pressure. In 95% of the patients in the control group, complete success could be observed; one patient had an IOP of 18 mmHg at the last follow-up visit. This patient required needling and open revision to reach the target pressure. The success rates did not significantly differ between the two groups (*p* = 0.568).

A total of 30% of eyes (six eyes) in the Healaflow^®^ group exhibited hypotony in the early postoperative phase (1–2 days after surgery), of which three eyes (15%) developed choroidal detachment (Table 1). One of these patients required intracameral Eyefill HD injection to stabilize the anterior chamber 2 days after surgery. Eye pressure increased afterwards, so there was no need for a second intervention. The number of hypotonous eyes decreased to 25% of eyes after 1 week. In the control group, 12% (three eyes) of patients developed hypotony, but none needed a surgical intervention. The number doubled to 24% (six eyes) after one week. Hypotony rates did not significantly differ at any time point (Table 1).

### 3.3. Postoperative Interventions (5-FU, Needling Rate)

In the Healaflow^®^ group, 19 of 20 eyes (95%) required 1.95 ± 0.83 5-FU injections, while in the control group, 24 of 25 (96%) patients required 1.48 ± 0.92 injections (*p* = 0.066). The achieved power was calculated to be 0.42.

In both groups, five patients (25% of patients in the Healaflow^®^ group, 20% of patients in the control group) required postoperative needling. Needling rates did not significantly differ at any time point (*p* = 0.731, Table 1). The achieved power was calculated to be 0.07.

### 3.4. Development of BCVA

There was no significant decline in BCVA in either group (Healaflow^®^ group: 0.57 ± 0.24, *p* = 0.4786, control group: 0.68 ± 0.26; *p* = 0.5481). The change in BCVA between the two groups was not significantly different (*p* = 0.1812). The achieved power was calculated to be 0.30.

## 4. Discussion

The present study focused on the early surgical outcomes of the implantation of a PreserFlo^®^ MicroShunt with MMC with and without the additional use of subtenon Healaflow^®^ in patients with POAG.

The use of Healaflow^®^, if injected during surgery with PreserFlo^®^ MicroShunt, proved to be safe and effective, with a comparable postoperative hypotony rate compared to the control group and a 88% complete success rate after 6 months.

Comparing both groups, the present study found that PreserFlo^®^ MicroShunt implantation with MMC was equally effective in terms of reduction in IOP and medication with or without the additional use of Healaflow^®^. Furthermore, 5-FU injection and needling rates did not significantly differ between the two groups (see above: *p* = 0.066 and *p* = 0.731).

So far, there have only been studies on the injection of Healaflow^®^ during filtrating surgeries without the use of implants. In these cases, Healaflow^®^ was injected as a spacer underneath the scleral flap and in some cases below the conjunctiva as well. In line with our results, the injection of Healaflow^®^ during trabeculectomy and deep sclerectomy was also found to be safe and effective [4,6,7,8].

There are only few case–control studies comparing the use of Healaflow^®^ to a control group, and there are conflicting data on the effect of Healaflow^®^ if used during trabeculectomy.

A recent study by Wu et al. indicated a favorable effect of the use of Healaflow^®^ regarding preservation of bleb morphology and reduction in final IOP. Wu and colleagues reported that the use of Healaflow^®^ during trabeculectomy (underneath the scleral flap and underneath the conjunctiva) led to a higher rate of functioning blebs and consequently lower IOP after 4–6 months [6]. Wang and colleagues, as well as Papaconstantinou et al., had a similar study design to ours, dividing patients into a Healaflow^®^ and “No Healaflow^®^” group. Both studies examined the effectiveness of Healaflow^®^ in trabeculectomy. Wang and colleagues were able to achieve a significantly greater reduction in IOP and medication over the long term through the use of Healaflow^®^ [9].

The results of the present study tend to corroborate the findings of Papaconstantinou et al., who were unable to demonstrate any significant difference in the reduction in pressure, medication, or rate of 5-FU through the use of Healaflow^®^ during trabeculectomy (underneath the scleral flap and underneath the conjunctiva), despite the fact that more Healaflow^®^ was injected here compared to the study by Wu et al. [7]. Nevertheless, it should be noted that in both studies, Healaflow^®^ was compared to a group that did not receive any anti-inflammatory agents during surgery. In the present implant-based study, the widely used MMC 0.2 mg/mL was applied and the additional benefit of Healaflow^®^ was evaluated against an MMC-treated group.

In the present study, Healaflow^®^ was injected subtenon after PreserFlo^®^ MicroShunt implantation in addition to MMC 0.2 mg/mL for 2 min before implantation. MMC is a DNA-crosslinking-alkylating agent that inhibits the cell cycle by intercalating DNA, thereby blocking cell proliferation [11]. Healaflow^®^, as a cross-linked hyaluronic acid, targets a different cellular mechanism: it is reported to prevent the accumulation of fibroblasts and the deposition of collagen without cytotoxicity [7,9]. Additionally, it is absorbed slowly by the tissue and is thought to have a space-occupying effect that minimizes the risk of over-filtration.

Mudhol et al. reported that Healaflow^®^ and low-dose MMC (0.1 mg/mL for 2 min), when used separately during trabeculectomy, are equally effective in preventing scarring regarding reduction in IOP, medication, and bleb formation after 12 months [11,12]. Mohamed et al. made similar observations, stating that trabeculectomy plus MMC at a dose of 0.3 mg/mL for 2 min and trabeculectomy plus Healaflow^®^ showed only non-significant differences in final IOP, success rates, and postoperative complications.

The present study’s data add that for filtrating surgery with an implant (Preserflo MicroShunt^®^), the effect on IOP, medication, and needling rate is not augmented when using both antifibrotic agents MMC in a 0.2 mg/mL concentration and Healaflow^®^ in the same approach. The herein observed needling rate did not significantly differ from the normal Preserflo Microshunt^®^ technique either in comparison with the current control cohort or with the rates reported in literature (*p* = 0.731) [13]. This might be because the additional effects of Healaflow^®^ on anti-scarring and maintaining space could be too minimal to cause significant differences in IOP, medication, or bleb formation between the two MMC-treated groups. It could be the case that MMC, due to its cytotoxic and aggressive effect on DNA replication and transcription and thus inhibition of fibroblast cell proliferation, has already reduced the inflammatory reaction to such an extent that an additional effect cannot be achieved with Healaflow^®^.

The question remains as to why Healaflow^®^ did not show any additional effect on IOP or needling rate in terms of its space-occupying effect. When comparing the present implant-based study to the currently available literature on trabeculectomies, Healaflow^®^ was injected subtenon on the one hand and beneath the flap on the other during trabeculectomy. It may be the case that the administration of Healaflow^®^ below the flap has the decisive effect on reducing intraocular pressure and long-term scarring. However, a larger number of cases is required to make definitive statements on this matter. Meanwhile, it would be interesting in future trials to compare the use of Healaflow^®^ only versus MMC in an implant-based trial, as bleb-associated complications caused by MMC such as atrophic blebs, scleral leakage, or melting [14] might be prevented using Healaflow^®^.

Due to its space-occupying effect and slow absorption, Healaflow^®^ was thought to prevent episodes of instable IOP. In the present cohorts, additional Healaflow^®^ did not prevent episodes of instable IOP, i.e., phases of hypotony; the total hypotony rates did not significantly differ between the two groups (see Table 1 for *p*-values). However, it was noticeable that the Healaflow^®^ group presented with earlier hypotony (peak at 1–2 days after surgery) than the control group (peak at one week after surgery). It could be argued that due to the described spacer effect of Healaflow^®^ hypotony could occur earlier than in the control group, where the primary bleb resistance must first be overcome for the bleb to form, leading to delayed hypotony. That is to say that the space created by Healaflow^®^ may, on the one hand, lead to the long-term resolution of adhesions and a well-formed prominent bleb, as described by Wang and colleagues [9]. On the other hand, this created space may initially lead to an increased outflow of aqueous humor caused by reduced tissue resistance, leading to a slightly higher hypotony rate in the beginning.

There is a non-significant trend towards higher IOP values at the 4–6 months timepoint in the Healaflow^®^ group. This could be attributed to one outlier who represents a patient presenting with an increase in IOP to 25 mmHg at this time point. This was a highly myopic POAG patient who had undergone a MIGS surgery (OMNI^®^ viscocanaloplasty ab interno) 8 months beforehand and presented with a high starting pressure of 63 mmHg. The patient required needling and open revision along with medication after PreserFlo^®^ MicroShunt insertion to achieve controlled IOP values. It could be argued that additional changes in the chamber angle led to this pronounced increase in pressure, which were harder to overcome.

Overall, more eyes in the Healaflow^®^ group had undergone prior surgery with MIGS (Table 1). However, all surgeries were performed more than eight months prior to the filtering procedure and were confined to the chamber angle, making a direct inflammatory effect on the bleb unlikely.

This study has several limitations. While the total number of eyes was considerable, studies involving a larger group of patients treated with Preserflo^®^ MicroShunt and using Healaflow^®^ are needed to better define the differences between the different surgical approaches. The sample size is certainly too small to draw any definite conclusions regarding development of IOP and medication. However, trends can already be detected in this first evaluation of the use of Healaflow^®^ in glaucoma surgery with implants. Due to its retrospective design, randomization was not feasible. Nevertheless, patients were assigned to the Healaflow or non-Healaflow group regardless of clinical variables such as age, IOP, severity of glaucoma, or conjunctival status in order to eliminate any bias. Although the follow-up of the present study is relatively short, ending after 6 months, it is questionable whether a longer time frame would have provided more relevant information on the effect of Healaflow^®^, given that Healaflow^®^ is likely to be absorbed during the first 3 weeks [7].

## 5. Conclusions

In conclusion, this is the first study evaluating the use of Healaflow^®^ in filtrating glaucoma surgery with the use of an implant. The use of Healaflow^®^, if injected during surgery with PreserFlo^®^ MicroShunt in addition to MMC, proved to be safe and effective. The sample size is too small to draw any definite conclusions; nevertheless, this first evaluation shows that intraocular pressure and the need for medication developed similarly up to 6 months after surgery in both scenarios (with or without the use of Healaflow^®^). Postoperative hypotony and needling rates did not significantly differ between the two groups (see Table 1 for *p*-values). The additional effects of Healaflow^®^ on anti-scarring and maintaining space could be too minimal to cause significant differences in IOP (*p* = 0.132 after 4–6 months) or medication (*p* = 0.118) when already treated with MMC.

## Figures and Tables

**Figure 1 jcm-14-06000-f001:**
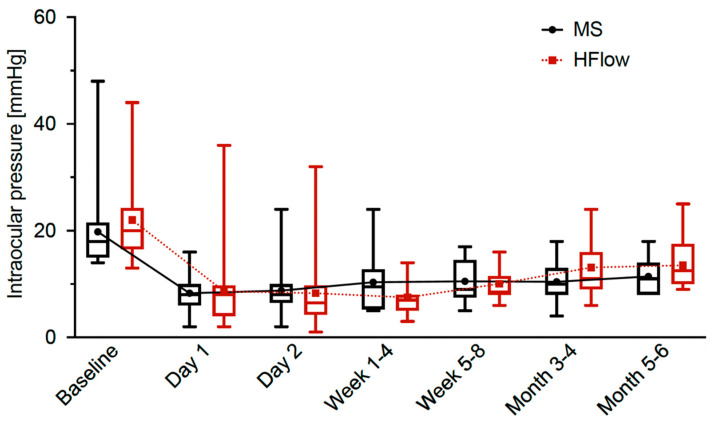
Boxplots showing intraocular pressure at each follow-up time point for PreserFlo^®^ MicroShunt implantation without (MS) and with Healaflow^®^ (HFlow). Mean IOP values are connected by a dotted line for each group. 95% CI for the differences in IOP at different time points: 1d −4.37 to 3.13, 2d −3.10 to 4.02, 1–4 weeks −3.57 to 4.27, 4–8 weeks −1.63 to 2.43, 3–4 months −4.26 to 0.70, 4–6 months −4.36 to 0.20.

**Figure 2 jcm-14-06000-f002:**
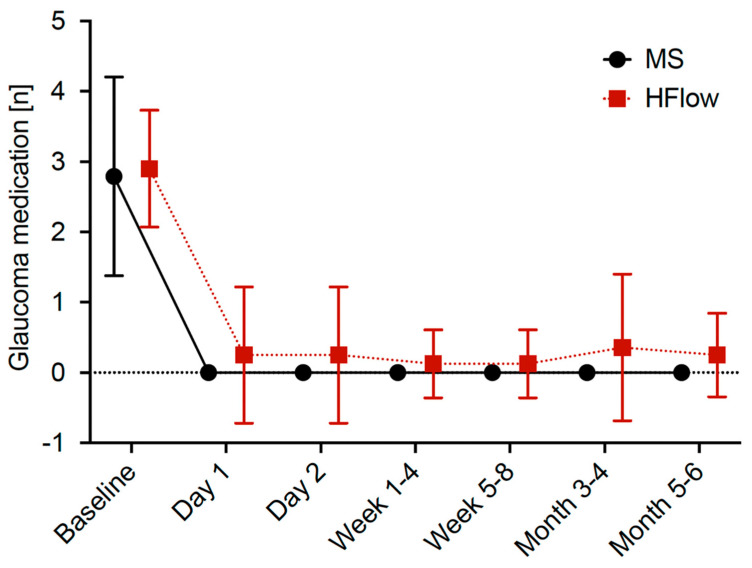
Medication. Line graph showing the medication at each time of follow-up for PreserFlo^®^ MicroShunt implantation without (MS) and with Healaflow^®^ (HFlow). The mean values of the numbers of medication are connected by a dotted line. 95% CI −0.54 to 0.04 at 5–6 months.

**Table 1 jcm-14-06000-t001:** Demographic and clinical characteristics of the Healaflow and control group; LIGS: less-invasive glaucoma surgery, MIGS: microinvasive glaucoma surgery, CPC: cyclophotocoagulation, SLT: selective lasertrabeculoplasty, MD: mean deviation.

	Healaflow Group (Mean ± SD; Range)	Control Group (Mean ± SD; Range)	*p*
No. of eyes (n)	20	25	
No. of patients (n)	20	25	
Gender (m/f)	8 m/12 f	18 m/7 f	*p* = 0.039
Mean age (y)	69.5 ± 11.8 (33–85)	69.5 ± 10.3 (37–86)	*p* = 0.808
Pseudophakia (n)	10	14	*p* = 0.769
Previous glaucoma surgeries/laser treatments (n)	8	4	*p* = 0.096
Trabeculectomy (n)	0	0	*p* = 1
Deep sclerectomy (n)	0	0	*p* = 1
LIGS (n)	0	1	*p* = 0.471
MIGS (n)	5	1	*p* = 0.041
CPC (n)	1	2	*p* = 1
SLT (n)	2	0	*p* = 0.221
>1 different procedure (n)	2	2	*p* = 1
Preoperative intraocular pressure (mmHg)	22.1 ± 8.5 (13–44)	19.8 ± 7.5 (14–48)	*p* = 0.341
Perimetry	−9.59 ± −8.79	−13.15 ± −13.6	*p* = 0.244
MD	(−26.56 to 0.38)	(−31.22 to 1.9)	
Hypotony rate (IOD ≤ 5 mmHg) (n)			
2 days after surgery	7 (35%)	3 (12%)	*p* = 0.083
+ choroidal detachment	3	1	*p* = 0.294
+ need for surgical intervention	1	0	*p* = 0.476
1–4 weeks after surgery	5 (25%)	6 (24%)	*p* = 1
+ choroidal detachment	0	0	*p* = 1
+ need for surgical intervention	0	0	*p* = 1
4–6 weeks after surgery	0	3 (12%)	*p* = 0.242
+ choroidal detachment	0	0	*p* = 1
+ need for surgical intervention	0	0	*p* = 1
3–4 months after surgery	0	0	*p* = 1
Needling rate (n)	5 (25%)	5 (20%)	*p* = 0.731
1–4 weeks after surgery	1	1	*p* = 1
4–6 weeks	1	2	*p* = 1
3–4 months	2	0	*p* = 0.221
5–6 months	1	2	*p* = 1

## Data Availability

The raw data supporting the conclusions of this article will be made available by the authors on request.

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
