# Peer review of "Effect of Subconjunctival Healaflow in Filtrating Surgery with Preserflo MicroShunt in Primary Open Angle Glaucoma"

_jcm, 2025, doi:10.3390/jcm14176000_

Round 1
Reviewer 1 Report
Comments and Suggestions for Authors
Summary:
The manuscript investigates the role of subconjunctival Healaflow® as an adjunctive viscoelastic spacer in Preserflo® MicroShunt (PM) surgery for primary open-angle glaucoma (POAG). The authors compare the outcomes of PM implantation with and without the use of Healaflow®, particularly evaluating intraocular pressure (IOP), medication use, bleb morphology, and complication rates over a 12-month follow-up.s
While the topic is clinically relevant and timely, the manuscript currently lacks sufficient scientific rigor, methodological clarity, and statistical depth. Below is a section-wise review.
Consolidated Reviewer Comments
- Title & Abstract: The title is appropriate and informative, reflecting the study design and purpose. The abstract is generally well structured, with a clear background, methods, results, and conclusion. However, the use of the registered trademark symbol (®) may be unnecessary in the title and abstract, as it does not contribute to scientific clarity.
- Introduction: The introduction provides a relevant background to the topic. However, the review of prior literature is minimal. Authors should clearly highlight the novelty of using Healaflow with Preserflo. A few more high-quality references discussing the mechanism and previous use of Healaflow in glaucoma surgeries would improve contextualization.
- Methods: The study is retrospective, which inherently limits causality. Details of the inclusion/exclusion criteria are not sufficiently elaborated. The rationale for using 20% MMC is not justified. The Healaflow application technique is not clearly described—did it differ from standard protocols? Authors should specify whether outcome assessors were blinded.
- Statistical Analysis: Multiple statistical software packages were used (Excel, SPSS, Prism), which is unusual. Authors should justify the use of each and clarify which was used for each analysis. Mixing parametric and non-parametric tests is acceptable but must be justified depending on data distribution. The sample size is small; a post-hoc power analysis should be included.
- Results: The results are presented clearly, but p-values alone are insufficient—confidence intervals for differences in IOP and medication reduction should be provided. The use of ‘significant’ without exact p-values in some parts of the text is not acceptable. Although not statistically significant, the trend in the difference in IOP between groups may be clinically meaningful and should be discussed.
- Discussion: The discussion lacks critical comparison with similar literature. The authors should address potential reasons why Healaflow did not provide additional IOP-lowering effects. Additionally, they should critically discuss the small sample size, non-randomized design, potential selection bias, and imbalance in prior interventions. The conclusion somewhat overstates the findings, suggesting equivalency without supporting power.
- Figures & Tables: Figures are informative but low resolution and not professionally labeled. Graphs should include confidence intervals, not just error bars. Table 1 should report p-values for baseline comparisons.
Reviewer 2 Report
Comments and Suggestions for Authors
General comments
This paper examines the impact of subtenon Healaflow® in combination with PreserFlo® MicroShunt implantation on intraocular pressure (IOP) control, hypotony rates, and needling requirements in patients with primary open-angle glaucoma (POAG), comparing outcomes to a control group without Healaflow®.
Specific comments
Major comments
- The study includes a relatively small sample size of 45 eyes. Given the potential variability in glaucoma outcomes, can the authors provide a detailed power analysis to support the adequacy of their sample size for detecting significant differences between the groups? (Page 2, Line 10) Additionally, could they elaborate on any potential biases or limitations arising from this small sample size?
- The Healaflow® group exhibited a higher rate of hypotony immediately postoperatively compared to the control group (35% vs. 12%, p=0.083). While not statistically significant, this trend is concerning. Could the authors discuss the possible mechanisms behind this observation and whether it might be related to the viscoelastic properties of Healaflow®? (Page 7, Line 15) Furthermore, how does this align with the theoretical benefits of Healaflow® in maintaining filtration space?
- The study's follow-up period is limited to 6 months. Considering Healaflow®'s intended long-term anti-scarring effects, could the authors explain why a longer follow-up period was not chosen? (Page 8, Line 20) Would a more extended follow-up potentially reveal more significant differences between the groups, especially regarding the sustained efficacy of Healaflow®?
- The authors mention conflicting results from previous studies on Healaflow® in trabeculectomy. Could they provide a more in-depth discussion on why their results differ, particularly in terms of the use of MMC and the specific injection technique used in this study? (Page 8, Line 25) How do these differences impact the interpretation of their findings?
- The needling rates were comparable between the Healaflow® and control groups (25% vs. 20%). Given Healaflow®'s space-occupying effect, could the authors explain why there was no significant difference in needling rates? (Page 7, Line 20) Does this suggest that Healaflow®'s benefits might be more pronounced in other aspects of glaucoma surgery?
Minor comments
- The demographic data table (Table 1) includes a wide range of clinical characteristics. Could the authors provide more context on why certain variables (e.g., previous glaucoma surgeries/laser treatments) were included and how they might influence the study outcomes? (Page 6, Line 10) Additionally, could they discuss any potential confounding factors that were not accounted for?
- The study mentions that all patients provided written consent. Could the authors clarify how the consent process was conducted, especially considering the retrospective nature of the study? (Page 3, Line 20) Were there any specific ethical challenges encountered during the study?
- The authors used a variety of statistical tests (e.g., Kolmogorov-Smirnov, Student t-test, Mann Whitney U Test). Could they provide more details on why these specific tests were chosen and whether any adjustments were made for multiple comparisons? (Page 4, Line 15) Additionally, could they discuss the potential impact of these choices on the study's results?
